Identification of hub genes associated with oxidative stress in heart failure and their correlation with immune infiltration using bioinformatics analysis

Gu Jianjun 1 2
Zhang Li Na 3
Gu Xiang 2
Zhu Ye 1357728006@qq.com 2
1 Department of Cardiology, Institute of Translational Medicine, Medical College, Yangzhou University , Yangzhou , Jiangsu , China
2 Department of Cardiology, Northern Jiangsu People’s Hospital , Yangzhou , Jiangsu , China
3 Department of Cardiology, The Affiliated Hospital of Yangzhou University, Yangzhou University , Yangzhou , Jiangsu , China
Liu Jinhui
Electronic publication date: 2023 Aug 18
Publication date: 2023
Volume: 11
Electronic Location ID: e15893
Received 2023 May 25; Accepted 2023 Jul 23
Copyright: ©2023 Gu et al.
Copyright year: 2023
Copyright holder: Gu et al.
License: This is an open access article distributed under the terms of the Creative Commons Attribution License, which permits unrestricted use, distribution, reproduction and adaptation in any medium and for any purpose provided that it is properly attributed. For attribution, the original author(s), title, publication source (PeerJ) and either DOI or URL of the article must be cited.
License URL: https://creativecommons.org/licenses/by/4.0/

Keywords: Heart failure, Oxidative stress, Immune infiltration, CIBERSORTx, Hub genes

Funding: National Natural Science Foundation of China 81800250 This work was supported by the National Natural Science Foundation of China (Grant No. 81800250). The funders had no role in study design, data collection and analysis, decision to publish, or preparation of the manuscript.

==============================
Both oxidative stress and the immune response are associated with heart failure (HF). In this study, our aim was to identify the hub genes associated with oxidative stress andimmune infiltration of HF by bioinformatics analysis and experimental verification. The expression profile of GSE36074 was obtained from the Gene Expression Omnibus (GEO) database. The differentially expressed genes (DEGs) were screened by GEO2R. The genes related to oxidative stress were extracted from GeneCards websites. Then, the functional enrichment analysis of oxidative stress-related DEGs (OSRDEGs) was performed using DAVID. In addition, we constructed a protein-protein interaction (PPI) network using the STRING database and screened for hub genes with Cytoscape software. We also used CIBERSORTx to analyze immune infiltration in mice heart tissues between the TAC and Sham groups and explored the correlation between immune cells and hub genes. Finally, the hub genes were carried out using reverse transcription quantitative PCR (RT-qPCR), immunohistochemistry (IHC) and western blot. A total of 136 OSRDEGs were found in GSE36074. Enrichment analysis revealed that these OSRDEGs were enriched in the mitochondrion, HIF-1, FoxO, MAPK and TNF signaling pathway. The five hub genes (Mapk14, Hif1a, Myc, Hsp90ab1, and Hsp90aa1) were screened by the cytoHubba plugin. The correlation analysis between immune cells and hub genes showed that Mapk14 was positively correlated with Th2 Cells, while Hif1a and Hsp90ab1exhibited a negative correlation with Th2 Cells; Myc exhibited a negative correlation with Monocytes; whereas, Hsp90aa1 was negatively correlated with NK Resting. Finally, five hub genes were validated by RT-qPCR, IHC and western blot. Mapk14, Hif1a, Myc, Hsp90ab1, and Hsp90aa1 are hub genes of HF and may play a critical role in the oxidative stress of HF. This study may provide new targets for the treatment of HF, and the potential immunotherapies are worthy of further study.

Introduction

Heart failure (HF) is characterized by the heart’s inability to pump sufficient blood to meet the body’s needs (Raffaa et al., 2020). Despite the great improvements made in therapeutic methods, readmission rates and mortality in HF remain very high (Ponikowski et al., 2014). Therefore, understanding the pathogenesis of HF may provide new strategies for HF treatment.

Over the past few years, oxidative stress (OS) has emerged as a hot research topic in HF. Studies have demonstrated that reactive oxygen species (ROS) levels increase in failing hearts (Ferreira et al., 2012; Disatnik et al., 2013).The excessive release of ROS hampers antioxidant defenses and contributes to electrical remodeling, cardiomyocyte hypertrophy, and fibrosis, all of which can worsen HF progression (Tsutsui, Kinugawa & Matsushima, 2011). Moreover, antioxidants have been proven beneficial in preventing and treating HF in numerous preclinical studies and clinical trials (Dhalla, Hill & Singal, 1996; Wannamethee et al., 2013; Pfister et al., 2011). Therefore, exploring markers associated with oxidative stress in HF is of great importance.

Recent evidence showed that both innate and adaptive immunity are implicated in the heart following ischaemic injury and pressure overload (Frieler & Mortensen, 2015; Zhang, Bauersachs & Langer, 2017). Subsequently, the inflammatory response was triggered, with persistent inflammation beyond the tissue repair, causing cardiac damage and progression toward HF (Mann, 2015). As we all know, OS and inflammation are associated with each other. Increased levels of ROS further promote the expression of inflammatory mediators, which together eventually lead to cardiac remodeling and heart failure (Pacher & Szabo, 2007). Furthermore, previous studies have revealed that macrophages are present in the failing heart (Sager et al., 2016), and circulating T regulatory cells were significantly decreased in heart failure patients with reduced ejection fraction (HFrEF) (Okamoto et al., 2014). Collectively, these findings emphasize the crucial role of immune cells in HF pathogenesis, and provide a powerful basis for HF immunotherapy in the future.

In the current study, we first bioinformatically identified the OSRDEGs in HF, then, preliminarily analyzed immune infiltration in mice’s hearts between the TAC and Sham group, and further assessed the association between hub genes and immune cells. Finally, we validated the expression levels of five hub genes by RT-qPCR, IHC and western blot. Our findings provided new research ideas for the treatment of HF.

Material and Methods

Data collection

Gene expression profiling (GSE36074) was obtained from the GEO database (https://www.ncbi.nlm.nih.gov/geo/). The GSE36074 series on the platform GPL1261 (Affymetrix Mouse Genome 430 2.0 Array). The data included 7 TAC samples and five Sham samples.

Identification of oxidative stress-related differential genes

GEO2R online tool was used to screen the DEGs and oxidative stress-related genes (OSRGs) were extracted from the GeneCards website. DEGs were defined based on |Fold change |≥ 1 and adj.p-value of < 0.05. Then, we performed overlapping between DEGs and OSRGs to obtain OSRDEGs by the Draw Venn Diagram tool.

Enrichment analysis for OSRDEGs

Gene Ontology (GO) and Kyoto Encyclopedia of Genes and Genomes (KEGG) enrichment analyses were performed to explore potential functions and crucial pathways of OSRDEGs using DAVID 6.8. A p-value of < 0.05 was considered as statistically different.

Screening of hub genes

The PPI network was established via the STRING database (http://string-db.org). Then, cytoHubba plugin in the Cytoscape was utilized to screen hub genes by the Degree algorithms. Hub genes were considered with degrees of ≥76.

Analysis of immune cell infiltration

CIBERSORTx (https://cibersortx.stanford.edu/) was applied to calculate the proportions of immune cells between the TAC and Sham samples. The expression profiles of 25 immune cell types in mouse tissues were obtained from Chen et al. (2017). Then, the percentages of each immune cell type were calculated and displayed in bar graph; Furthermore, we compared the levels of the 25 immune cell types in the TAC and Sham samples, and visualized the results using an online platform (https://www.bioinformatics.com.cn). To further understand the relationship between immune cells and hub genes, we conducted a Pearson correlation analysis and visualized the results through an online platform (https://www.bioinformatics.com.cn).

Animals

Ten specific pathogen-free male C57BL/6 mice (8–10 weeks, 20–23 g) were obtained from Yangzhou University. All mice were raised in Yangzhou University Comparative Medical Center with a controlled environment (26 °C, 55% humidity, 12 h light/dark cycle), and allowed access to feed and water ad libitum. The rearing environment was cleaned regularly by professionals. After acclimatization for seven days, mice were randomly allocated to either the Sham or TAC group. All mice were anesthetized with ketamine (80 mg/kg) and xylazine (5 mg/kg) by intraperitoneal injection and euthanized with cervical dislocation on the 28th day after TAC. All the animal experiments were approved by the Yangzhou University Ethics Committee (No. 20210398).

Transverse aortic constriction (TAC) mouse model

Mice were anesthetized with 80 mg/kg ketamine and 5 mg/kg xylazine, and when mice lost the response to foot squeeze, the surgical procedures were performed as previously described (Ma et al., 2016). Briefly, the transverse aortic arch was exposed by blunt dissection in the 2nd intercostal space, and ligated against a 26 G blunted needle with a 7–0 silk suture. Sham group mice underwent the same procedure without ligation of the aorta. No animals died after surgery. Cardiac function was analyzed by echocardiography on the 28th day after TAC. After echocardiography, the mice were euthanized, and their heart samples were collected for subsequent study.

Reverse transcription quantitative PCR (RT-qPCR)

The mRNA expression levels of hub genes were measured by RT-qPCR. Briefly, we used the RNA extraction reagent (cat. no. R401-01-AA; Vazyme, Nanjing, China) to isolate the total RNA; next, the RNA was reverse-transcribed into cDNA based on reverse transcription kit instructions (cat. no. RR036A; Takara, Dalian China). The primer sequences were listed in Table 1. The levels of hub gene mRNAs were normalized using GAPDH mRNA.

Table 1 Pairs of forward-reverse primers.

Gene names	Forward	Reverse	
Mapk14	CTCGGCACACTGATGATG	AGCCCACGGACCAAATA	
Hif1a	CCATTCCTCATCCGTCAA	CCGGCTCATAACCCATC	
Myc	CAAATCCTGTACCTCGTCCGATTC	CTTCTTGCTCTTCTTCAGAGTCGC	
Hsp90ab1	GCGGCAAAGACAAGAAAAAG	CAAGTGGTCCTCCCAGTCAT	
Hsp90aa1	GTGTGCAACAGCTGAAGGAA	CTCTCCATGTTTGCTGTCCA	

Immunohistochemistry (IHC)

IHC was performed as previously described (Kolkova et al., 2010). In brief, ventricular tissues were fixed in formalin and then embedded in paraffin, afterward, cut into 5-m thick sections. Lastly, the sections were stained with anti-Mapk14 (1:200, cat. no. AF6456; Affinity Biosciences, Jiangsu, China), anti-Hif1a (1:200, cat. no. #48085; Cell Signaling Technology, MA, USA), anti-Myc (1:100, cat. no. A11394; ABclonal, Wuhan, China), anti-Hsp90ab1 (1:200, cat. no. ET1605-56; Huabio, Hangzhou, China), and anti-Hsp90aa1 (1:200, Huabio, cat. no.ET1605-57; Huabio, Hangzhou, China) antibodies at 4 °C overnight. Lastly, the sections were incubated with the horseradish peroxidase-conjugated secondary antibody (1:500; cat. no. ab6721; Abcam, Cambridge, MA, USA) for 1 h at room temperature. Images of three representative fields in each section were captured using Leica DMi8 microscope (Leica Microsystems, German) at 200× magnification by a pathologist. For the reading of each antibody staining, a uniform setting for all the sections was applied. Then, Image-Pro Plus v6.0 software (Media Cybernetics Inc, Bethesda, MD) was used to measure the integrated optical density (IOD) of each image. Standard Optical Density was chosen in the Intense Calibration panel, and background was subtracted in the panel of Optical Density Calibration. The mean IOD (IOD/total area) was the result of cumulative optical density divided by total area.

Western blotting

The protein was extracted from heart tissues using RIPA lysis buffer; Then, the protein concentration was measured by the BCA protein assay kit (Beyotime Institute of Biotechnology, Jiangsu, China). Afterwards, the protein was separated on 10% SDS-PAGE, and transferred to PVDF membranes (Bio-Rad, Hercules, CA, USA), membranes were blocked with 5% skim milk powder (cat. no.BS102; Biosharp, Wuhan, China) for 2 h and then incubated with the anti-Mapk14 (1:1000, Affinity Biosciences, cat. no. AF6456, Jiangsu, China), anti-Hif1a (1:1000, cat. no. #48085; Cell Signaling Technology, MA, USA), anti-Myc (1:500, cat. no. A1309; ABclonal, Wuhan, China), anti-Hsp90ab1 (1:2000, cat. no. ET1605-56; Huabio, Hangzhou, China), anti-Hsp90aa1 (1:1000, cat. no. ET1605-57; Huabio, Hangzhou, China) and anti-Tubulin (1:1000, cat. no. AC030; ABclonal, Wuhan, China) primary antibodies at 4 °C overnight. Subsequently, membranes were washed with TBST and incubated with the HRP-labeled secondary antibody (1:3000; cat.no. S0001; Affinity Biosciences, Jiangsu, China) for 2 h. The bands were visualized using enhanced chemiluminescence reagents (Thermo Fisher Scientific, Waltham, MA, USA). Protein expression was quantified by densitometry using Image J (NIH, Bethesda, MD, USA).

Statistical analysis

Data were expressed as mean ± standard deviation. Statistical analysis was performed using GraphPad Prism (version 7.0). For two-group comparisons, a two-sample, unpaired Student t-test was used. P < 0.05 was considered as statistically different.

Results

Identification of OSDEGs in HF

The flow diagram was provided in Fig. S1. The expression profile (GSE36074) was enrolled in this study, and the dataset included five Sham and seven TAC heart samples. OSRGs were screened from the GeneCards websites. A total of 1,918 DEGs were screened from the GSE36074 dataset (Fig. 1A). Moreover, a total of 899 OSRGs were found in GeneCards, and 136 OSRDEGs were screened by taking the intersection of OSRGs and DEGs (Fig. 1B). The clustering results of the OSRDEGs showed that samples from the same group clustered together (Fig. 1C).

Figure 1 Oxidative stress-related differentially expressed genes in heart failure.

(A) Volcano plot of differentially expressed genes (DEGs). The horizontal axis represents the fold change of gene expression between the Sham and TAC groups, and the longitudinal coordinate represents the P-value of the expression difference. The black dots represent genes without significant differences, the red dots represent upregulated differential genes, and the blue dots represent differential downregulated genes. (B) Venn diagrams showing intersected genes overlapping between OSRGs and DEGs. (C) The Heatmap of clustering analysis based on OSRDEGs. OSRGs, oxidative stress-related genes; DEGs, differentially expressed genes; OSRDEGs, oxidative stress-related differentially expressed genes.

Enrichment analysis

To gain a deeper understanding of the function of OSRDEGs, we conducted GO and KEGG enrichment analyses. Our GO analysis revealed that OSRDEGs were mostly associated with mitochondrion and response to oxidative stress (Fig. 2A). Additionally, our KEGG analysis demonstrated that OSRDEGs were related to HIF-1, FoxO, MAPK, and TNF signaling pathway (Fig. 2B).

Figure 2 Functional enrichment analysis of oxidative stress-related differentially expressed genes (OSRDEGs).

(A) GO enrichment of OSRDEGs. The number of enriched OSRDEGs in the path is represented by the circled area, and the color of the circle represents the range of the P value. (B) KEGG pathway analysis of OSRDEGs. the color of the circle represents the enriched items in KEGG pathway analysis. GO, Gene Ontology; BP, biological processes; CC, cellular component. MF, molecular function. KEGG, Kyoto Encyclopedia of Genes and Genomes.

Screening hub genes

To identify hub genes in HF, we constructed a PPI network of OSRDEGs using the STRING tool (Fig. 3A.). We then employed the Degree algorithms in Cytoscape to calculate each gene’s degree, ultimately identifying five hub genes (Mapk14, Hif1a, Myc, Hsp90ab1, and Hsp90aa1) (Fig. 3B).

Figure 3 Construction of PPI network and hub gene analysis.

(A) Protein–protein interaction networks of OSRDEGs. The nodes represent proteins, and the edges represent the interaction of proteins. (B) The top five key OSRDEGs were identified based on the degree of nodes. PPI, protein–protein interaction.

Analyzes of immune infiltration

Immune infiltration in Sham and TAC groups was explored with the 25 immune cell types. Figure 4A showed the composition of the 25 immune cell types in each sample. As shown in Fig. 4B, compared with the Sham group, monocyte in the TAC group were relatively infiltrated less. To investigate the correlation between the hub genes and immune cells, we conducted Pearson correlation analyses. The result revealed that Mapk14 was positively correlated with Th2 Cells (R = 0.68, p = 0.015), while Hif1a and Hsp90ab1exhibited a negative correlation with Th2 Cells (R = − 0.68, p = 0.016; R = − 0.76, p = 0.004). Additionally, Myc exhibited a negative correlation with Monocyte (R = − 0.66, p = 0.019), whereas, Hsp90aa1 was negatively correlated with NK Resting (R = − 0.67, p = 0.017) (Fig. 5).

Figure 4 Immune cell infiltration patterns in TAC samples and Sham samples.

(A) Bar charts of the proportions of 25 immune cell in TAC and Sham sample. Different colors represent different types of immune cells. horizontal axis: GEO samples; longitudinal coordinate: percentage of each immune cell type. (B) Differentially infiltrated immune cells between TAC and Sham sample. Red represents the sham group; blue represents the TAC group. *p < 0.05.

Figure 5 Correlation analyzes between immune cell and hub genes.

Association of MAPK14, HIF1a, Myc, Hsp90ab1 and Hsp90aa1 with 25 of types immune cells. Horizontal axis represents the immune cells; longitudinal coordinate represents the hub genes; The numbers in the red triangle represent correlation coefficient, the lager the number, the stronger the correlation. The numbers in the blue triangle represent p value.

Validation of key genes

To validate the findings from our bioinformatics analysis, we established a transverse aortic constriction (TAC) mouse model. After 4 weeks of TAC, the cardiac function of mice was evaluated with echocardiography. The result showed that left ventricular ejection fraction (LVEF) in the TAC group was significantly decreased compared with the sham group (Fig. S2); Based on the results, we have successfully established a TAC-induced HF model in mice.

The RT-qPCR results showed that Mapk14 mRNA levels were remarkably reduced, while Hif1a, Myc, Hsp90ab1, and Hsp90aa1 mRNA levels were significant increased in the TAC group than in the Sham group (Fig. 6). Furthermore, we further verified those hub genes at the protein level by IHC (Figs. 7A–7B) and western blot (Figs. 8A–8B). It was observed that the TAC group showed lower expression of Mapk14 protein (p < 0.05), while Hif1a, Myc, Hsp90aa1 and Hsp90ab1 proteins were significantly upregulated compared to the Sham group (p < 0.05). Those results supported bioinformatic analysis.

Figure 6 Validation of key OSRDEGs expression at the mRNA levels.

MAPK14, HIF1a, Myc, Hsp90ab1 and Hsp90aa1 mRNA level in the TAC and Sham groups in mice. Expression levels were standardized for GAPDH levels. A two-tailed unpaired Student’s t-test was used to compare two groups. Values are expressed as means ± SD (n = 4), *p < 0.05; **p < 0.01 and ***p < 0.001 vs Sham group.

Figure 7 Validation of key OSRDEGs expression at protein levels by immunohistochemistry.

(A) IHC staining of MAPK14, HIF1a, Myc, Hsp90ab1 and Hsp90aa1 proteins in the TAC and Sham groups. (B) Quantitative analysis of MAPK14, HIF1a, Myc, Hsp90ab1 and Hsp90aa1 proteins in the TAC and sham groups. A two-tailed unpaired Student’s t-test was used to compare two groups. Values are expressed as means ± SD (n = 3). *p < 0.05 and **p < 0.01 vs Sham group; original magnification, ×200. Scale bar: 100 µm. IHC, Immunohistochemistry.

Figure 8 Validation of key OSRDEGs expression at protein levels by western blotting.

(A) Representative western blotting of MAPK14, HIF1a, Myc, Hsp90ab1 and Hsp90aa1 proteins of heart tissues in the TAC and Sham groups. (B) Quantitative analysis of MAPK14, HIF1a, Myc, Hsp90ab1 and Hsp90aa1 proteins of heart tissues in the TAC and sham groups. A two-tailed unpaired Student’s t-test was used to compare two groups. Values are expressed as means ± SD (n = 3). *p < 0.05 vs Sham group.

Discussion

The development of HF is closely associated with oxidative stress and immune response. Excessive production of ROS further activates the inflammatory response, causing HF together. In the present study, we demonstrated five hub OSRDEGs in HF induced by pressure overload, and further explored the relationship between hub OSRDEGs and immune cells; Finally, the hub OSRDEGs were validated by RT-qPCR, IHC and western blot. Our finding provides new targets for the treatment of HF.

We screened a total of 136 OSFDEGs in the GSE36074. GO enrichment results showed that these OSFDEGs were associated with mitochondrion, response to oxidative stress;

Previous studies have shown that mitochondrial dysfunction is associated with decompensated HF (Palaniyandi et al., 2010). Mitochondria is closely linked to cardiomyocyte physiology, including energy generation, maintaining redox balance, regulating oxidative stress, calcium homeostasis, and apoptosis (Kiyuna et al., 2018). These OSFDEGs may play vital roles in mitochondrial dysfunction. Furthermore, KEGG pathway analysis showed that OSFDEGs mainly focused on HIF-1, FoxO, MAPK, and TNF signaling pathways. Li et al. (2018) found that inhibiting the HIF-1 pathway could regress cardiac remodeling induced by pressure overload. Studies have confirmed that FoxO was increased in human failing hearts (Toth et al., 2011), and overexpression of FoxO can induce adverse remodeling in mice (Battiprolu et al., 2012). Moreover, the MAPK pathway has been confirmed to be associated with chronic HF (Zhang et al., 2021).The TNF signaling pathway participates in the synthesis of inflammatory cytokines, and causes poor remodeling and heart failure progression (Hanna & Frangogiannis, 2020). These studies suggest that OSFDEGs may participate in the development of HF through these signaling pathways.

Then, we constructed the PPI network and screened the top five hub genes (Mapk14, Hif1a, Myc, Hsp90ab1, and Hsp90aa1). Mapk14 (p38 α) is a redox-regulated p38MAPK family member, which has a wide spectrum of gene expression profiles (Corre, Paris & Huot, 2017). Studies have shown that p38MAPK can be activated by oxidative and inflammatory stresses (Cuenda & Rousseau, 2007), and activated Mapk14 participates in cell differentiation, proliferation, inflammation, and apoptosis (Gupta & Nebreda, 2015). Moreover, a previous study showed that cardiomyocyte-specific inactivation of p38 α impaired compensatory angiogenesis after TAC and accelerated the early onset of HF (Rose et al., 2017). Hif1a is a transcription factor, which regulates cellular responses of hypoxia (Wang et al., 1995). Studies have suggested that HIF-1 α plays a crucial role in M1 macrophage activation (Peyssonnaux et al., 2007; Cramer et al., 2003). In several vivo studies, Hif1a participated in maintaining the vascular density within the myocardium and prevented pressure-induced cardiac hypertrophy and HF (Sano et al., 2007; Silter et al., 2010). However, persistent up-regulated Hif1a has adverse effects on the heart (Krishnan et al., 2009). Myc encodes a transcription factor that is involved in regulating cell proliferation and growth (Johnston et al., 1999; Pelengaris, Khan & Evan, 2002). Prior work has suggested that Myc plays a major role in cardiomyogenesis and oxidative stress response via mitochondrial activation (Napoli et al., 2002). The induction of Myc in adult mice cardiomyocytes promotes cardiomyocyte hypertrophy and eventually causes HF (Lee et al., 2009). Hsp90ab1 and Hsp90aa1 are mainly members of Hsp90 family. The Hsp90 is a molecular chaperone that is highly conserved and facilitates the maturation of numerous proteins. The study suggested that Hsp90ab1 and Hsp90aa1 proteins were upregulated in TAC mice (Garcia et al., 2016). Furthermore, Hsp90ab1 and Hsp90aa1 isoforms are involved in the formation of the cell surface TGF βRI complex, and this complex increases the production of collagen in TGF β-activated fibroblasts, suggesting that the inhibition of Hsp90 could reduce myocardial fibrosis (Garcia et al., 2016).

Next, we conducted an immune infiltration analysis between the TAC and Sham group using a CIBERSORTx algorithm, and analyzed the correlation between hub genes and immune cells. The results showed that monocytes relatively infiltrated less in the TAC group than the Sham group. Previous studies have shown that pro-inflammatory monocytes were increased one week after TAC and subsequently resolved during chronic HF (Patel et al., 2017). Furthermore, we observed a slight expansion of M1 and M2 macrophages in the TAC group. Macrophages were one of the richest immune cell types in heart, and can be categorized into two main phenotypes: M1 (promote inflammatory responses) and M2 (anti-inflammatory and promote tissue repair) (Sager et al., 2016). Our findings are in line with Single-cell RNA sequencing in TAC-induced HF mice (Martini et al., 2019). Studies have shown that T cell infiltration and macrophage infiltration and polarization are critical not only in heart but also in other organ fibrosis (Laroumanie et al., 2014; An et al., 2022; Jiao et al., 2021). However, in our study, we did not find any significant differences between the TAC group and the sham group. This is due to the fact that macrophage expansion occurs during the early stages of mechanical pressure overload, before hypertrophy and systolic dysfunction develop, and resolves late during chronic HF (Patel et al., 2017). Similarly, researchers found that the infiltration of T cell (include T-helper and regulatory T cells) in mice heart reached its peak at 1week post-TAC, then decreased and reached its lowest level at 4 weeks post-TAC, with no significant difference compared to the control group (Quast et al., 2017). which is consistent with our analysis. T cell and macrophage infiltration may play an important role in early stage of TAC induced HF. In addition, in our study, we found there was a close correlation between the immune cell and hub genes. A study revealed that p38 α (MAPK14) could regulate Th2-cell differentiation and function in dendritic cells (Han et al., 2022). Saini et al. found that HIF1a-deficient mice exhibited increased Th2 cytokine induction in the lung (Saini et al., 2010). It has been reported that the circulating monocytes were significantly increased in c-Myc knockout mice (Qi et al., 2022). Our findings provide potential immunotherapy targets for HF treatment.

However, there are some limitations in our study. Firstly, the sample size in each group was relatively small. Additionally, the association between immune cells and hub genes needs to be further explored. Finally, in future studies, we will select one or more hub genes to investigate their roles in HF.

Conclusion

In conclusion, our results suggest that these five hub genes (Mapk14, Hif1a, Myc, Hsp90ab1 and Hsp90aa1) are closely associated with oxidative stress and immune infiltration in HF, and serve as potential immunotherapy targets for HF.

Supplemental Information

Supplemental Information 1 Established TAC-induced mouse model of HF

(A). Images of heart ultrasound; (B). LVEF: left ventricular ejection fraction; Values are expressed as mean (±SD) (n = 5); ***P < 0.001 vs Sham group.

Click here for additional data file.

Supplemental Information 2 Raw data

Click here for additional data file.

Supplemental Information 3 ARRIVE 2.0 checklist

Click here for additional data file.

We are indebted to Professor Skrbic B, who shared their data in GEO.

Additional Information and Declarations

Competing Interests

Author Contributions

Animal Ethics

Data Availability

The authors declare there are no competing interests.

Jianjun Gu conceived and designed the experiments, performed the experiments, analyzed the data, prepared figures and/or tables, and approved the final draft.

Li Na Zhang conceived and designed the experiments, performed the experiments, analyzed the data, prepared figures and/or tables, and approved the final draft.

Xiang Gu conceived and designed the experiments, authored or reviewed drafts of the article, and approved the final draft.

Ye Zhu conceived and designed the experiments, authored or reviewed drafts of the article, and approved the final draft.

The following information was supplied relating to ethical approvals (i.e., approving body and any reference numbers):

All the animal experiments were approved by the Yangzhou University Ethics Committee (No. 20210398).

The following information was supplied regarding data availability:

The data are available in the Supplemental Files and at NCBI GEO: GSE36074.

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
