# Peer review of "Identification of hub genes associated with oxidative stress in heart failure and their correlation with immune infiltration using bioinformatics analysis"

_PeerJ, doi:10.7717/peerj.15893_

## Round 0.1 · original submission · Major Revisions

Authors should revise according to the suggestions of reviewers. The modifications should be marked. A point to point response letter is needed.

Reviewer 1 ·

Basic reporting

In the manuscript entitled “Identification of hub genes associated with oxidative stress in heart failure and their correlation with immune infiltration using bioinformatics analysis”, the authors employed several bioinformatics analysis strategies to find that Mapk14, Hif1a, Myc, Hsp90ab1, Hsp90aa1 play important roles in oxidative stress of heart failure. Then, the authors validated their finding in TAC mice model. In short, it’s an interesting study. However, there are some issues should be addressed:

1. It would be better if the authors can add Western Blot results to validate protein expression levels of Mapk14, Hif1a, Myc, Hsp90ab1, Hsp90aa1 in their mice model.
2. There are many studies showed that T cell infiltration and macrophage infiltration and polarization are critical not only in heart but also in other organ fibrosis. (J Cell Physiol. 2022 Jan;237(1):983-991)( Front Immunol. 2021 Aug 26;12:735014) (Cells. 2021 Nov 6;10(11):3057) (Br J Pharmacol. 2023 Apr 19. doi: 10.1111/bph.16096.). However, it looks like only monocyte fraction significantly decreased after TAC in Figure 5. It would be better if the authors can discuss in detail.

Experimental design

1. It would be better if the authors can illustrate how they identity HIF-1, FaxO, MAPK and TNF signaling pathways in Figure 3B since it looks like all signaling pathways showed in Figure 3B have small p values.
2. The authors should add more information in their figure legends to let audience know how to read their figures.

Validity of the findings

no comment

·

Basic reporting

In the manuscript “Identification of hub genes associated with oxidative stress in heart failure and their correlation with immune infiltration using bioinformatics analysis”, Gu et al identify the hub genes associated with oxidative stress and immune infiltration of HF by bioinformatics analysis and experimental verification. This study conducted an enrichment analysis and found that the differentially expressed genes in oxidative stress responses (OSRDEGs) are concentrated in the mitochondrion, HIF-1, FoxO, MAPK, and TNF signaling pathways. Five primary genes were identified (Mapk14, Hif1a, Myc, Hsp90ab1, and Hsp90aa1) using the cytoHubba plugin. The correlation between immune cells and these hub genes was also examined. Mapk14 had a positive correlation with Th2 Cells, while Hif1a and Hsp90ab1 showed a negative correlation with Th2 Cells. Myc had a negative correlation with Monocytes, and Hsp90aa1 was negatively correlated with NK Resting. The five primary genes were validated using RT-qPCR and IHC techniques. The study suggested that these five genes (Mapk14, Hif1a, Myc, Hsp90ab1, and Hsp90aa1) are crucial in heart failure (HF), particularly in its oxidative stress responses. The topic is interesting, and this research offers potential new therapeutic targets for HF treatment and suggests that the role of potential immunotherapies should be explored further. There are several comments should be addressed before considering publication.
1. The relevance and detail provided by Figure 1 are limited and, therefore, it would be more appropriately placed within the supplementary data section of the paper.
2. The methods section currently lacks the necessary detail. For increased clarity and replicability, the authors should provide details information about the chemical reagents, antibodies, and kits used in the experiments. Specific details such as the names, sources, and catalog numbers of these items would be particularly useful and should be included.
3. What were the methods of quantification for the immunohistochemistry results? Please include detailed information about the specific procedures and the software used in the manuscript.
4. In Figure 2C, could you please verify the color indicator bar? Does it range from 1 to 0?
5. In Figure 8A, the clarity of the IHC staining image is insufficient and some signals appear questionable. Providing a clearer image would improve understanding, and it would be beneficial if the authors could also supply a magnified version of the picture for better visualization.
6. The figure legends in the current version of the manuscript are too simplistic and lack necessary detail. It would greatly improve the clarity and comprehensiveness of the paper if the authors could provide more detailed descriptions in these legends.
7. There are a number of grammatical errors in this manuscript; please review and revise it carefully.

Experimental design

no comment

Validity of the findings

no comment

Additional comments

no comment

---

## Round 0.2 · accepted · Accept

The authors have addressed the reviewers' concerns properly and revised the manuscript accordingly. The manuscript can be accepted for publication in its current form.

Reviewer 1 ·

Basic reporting

No comment.

Experimental design

No comment.

Validity of the findings

No comment.

·

Basic reporting

Thanks for the authors reply, all of my comments and concerns have been sucessfully addressed, now I have not any other comments.

Experimental design

None

Validity of the findings

None

Additional comments

None